# A key region of molecular specificity orchestrates unique ephrin-B1 utilization by Cedar virus

Rhys Pryce[1],* , Kristopher Azarm[2],* , Ilona Rissanen[1,3], Karl Harlos[1], Thomas A Bowden[1] , Benhur Lee[2,4]

The emergent zoonotic henipaviruses, Hendra, and Nipah are responsible for frequent and fatal disease outbreaks in domestic animals and humans. Specificity of henipavirus attachment glycoproteins (G) for highly species-conserved ephrin ligands underpins their broad host range and is associated with systemic and neurological disease pathologies. Here, we demonstrate that Cedar virus (CedV)—a related henipavirus that is ostensibly nonpathogenic—possesses an idiosyncratic entry receptor repertoire that includes the common henipaviral receptor, ephrin-B2, but, distinct from pathogenic henipaviruses, does not include ephrin-B3. Uniquely among known henipaviruses, CedV can use ephrin-B1 for cellular entry. Structural analyses of CedV-G reveal a key region of molecular specificity that directs ephrin-B1 utilization, while preserving a universal mode of ephrin-B2 recognition. The structural and functional insights presented uncover diversity within the known henipavirus receptor repertoire and suggest that only modest structural changes may be required to modulate receptor specificities within this group of lethal human pathogens.

## Introduction

The prototypic henipaviruses (HNVs), Hendra virus (HeV), and Nipah (NiV) virus are biosafety level four (BSL4) pathogens responsible for severe human disease that is associated with rapid onset and case fatality rates that can exceed 90% (1, 2, 3). In addition, the extreme disease pathologies, absence of a licensed vaccine, paucity of medical intervention options, and zoonotic potential delineate HNVs as an acute and persistent threat to global biosecurity, economy, and health (4). Although zoonotic spillover is typically associated with transmission from chiropteran reservoirs, or via infection of domestic animal intermediates, such as pigs and horses, transmission is not restricted to cross-species spillover events. Direct human-to-human spread is frequent and highlights the pandemic potential of HNVs (5, 6, 7).

Serological studies suggest that HNVs occupy a broad geographic range coincident with, but not restricted to, the home range of reservoir bat species of the order *Chiroptera* (8, 9). Although there is evidence for the existence and spillover of previously uncharacterized HNVs in Africa and Central- and South-America (10, 11, 12, 13, 14, 15), an accurate appraisal of the human impact of such HNVs is likely hindered by the spectrum of clinical outcomes inherent to diverse HNV species (16). Indeed, the indirect association of a novel HNV, Mójiāng virus (MojV), with the death of three miners in China highlights the potential of non-chiropteran hosts as reservoirs of lethal HNVs (17). The putatively rat-borne MojV uses a structurally divergent attachment glycoprotein to mediate a distinct host-cell entry pathway (18). The continued discovery and emergence of novel HNV species underscores the indeterminate global health threat that they pose (4, 12).

Cedar virus (CedV) is a *Henipavirus* species isolated from the excreta of *Pteropus* bat colonies in Queensland, Australia (19). Although geographically, genetically, and serologically related to the highly virulent prototypic HNVs, CedV is apathogenic in small animal models (19). The stark disparity in HNV pathogenesis has been attributed, in part, to the lack of an otherwise conserved RNA-editing site and the alternate reading frame coding capacity for accessory proteins within the CedV phosphoprotein (P) gene (19). In NiV and HeV, RNA editing facilitates the production of the accessory proteins, V and W, which are capable of antagonizing the IFN response. The absence of these immunomodulatory accessory proteins in CedV results in a failure to mitigate the antiviral effects of the type I IFN response and likely represents a critical factor in determining infection outcomes (20).

The single-stranded negative-sense RNA genome of HNVs encodes two surface glycoproteins: the receptor-binding glycoprotein (G) and the type I viral fusion protein (F), which work in concert to orchestrate cellular entry (21, 22, 23). Binding of the HNV-G to cell surface receptors belonging to the ephrin ligand family initiates pH-independent activation of F, triggering a fusion cascade that results in the ultimate merger of viral and cellular membranes. HNV-G proteins comprise a short N-terminal cytosolic region, single-pass transmembrane domain, oligomerization-mediating and

[1]Division of Structural Biology, Wellcome Centre for Human Genetics, University of Oxford, Roosevelt Drive, Oxford, UK    [2]Icahn School of Medicine at Mount Sinai, New York, NY, USA    [3]Helsinki Institute for Life Science, University of Helsinki, Biocenter 3, Helsinki, Finland    [4]Global Virus Network Center of Excellence, Center for Virology, Icahn School of Medicine at Mount Sinai, New York, NY, USA

Correspondence: thomas.bowden@strubi.ox.ac.uk; benhur.lee@mssm.edu
*Rhys Pryce and Kristopher Azarm contributed equally to this work

fusion-activating stalk region, and C-terminal receptor-binding β-propeller domain ([24], [25]). Orthologs of the two established HNV receptors, ephrin-B2, and ephrin-B3 are extremely well conserved across numerous reservoir and vector species and are recognized by all known ephrin-tropic HNV-G proteins with a conserved binding mode ([26], [27], [28]). Utilization of ephrins as cellular entry receptors is fundamental to the broad cell type and species tropism of HNVs and underscores key features of HNV zoonosis and pathogenesis ([29], [30]).

Despite lacking canonical type I IFN antagonists (V and W) ([19], [20]), CedV does possess a functional C accessory protein, the counterparts of which exhibit type I IFN antagonism in NiV ([31], [32], [33], [34]). Furthermore, CedV is able to establish a productive, albeit self-limiting, infection in Syrian hamsters that is more robust when inoculated via the intranasal versus intraperitoneal route ([20]). Together, these observations suggest that the identity of cellular receptors for CedV and the efficiency with which they are used may constitute additional modifiers of pathogenicity. Here, we sought to delineate the functional entry receptor repertoire of CedV and elucidate the molecular determinants of receptor specificity. In our integrated structural and functional analysis, we demonstrate that in addition to using the common HNV receptor, ephrin-B2, CedV uses ephrin-B1, a receptor with no precedent of HNV usage. Structural analyses reveal that whereas CedV-G conforms to the established mode of HNV-G mediated ephrin receptor engagement, subtle structural features of the glycoprotein contribute to its unique ephrin ligand specificity. These data highlight functional diversity amongst HNV-G proteins and provide mechanistic insight into potential modulators of HNV pathobiology.

# Results

## The crystal structure of CedV-G reveals a conserved receptor-binding architecture

Although reported to use the common entry receptor, ephrin-B2 ([19], [35]), CedV-G is genetically distinct from all characterized ephrin-tropic HNVs (26%, 28%, and 31% identical to GhV-G [Ghana virus], HeV-G, and NiV-G, respectively) ([27]). To assess the extent to which this sequence divergence is reflected at a structural level, we determined the crystal structure of the CedV-G receptor-binding domain to 2.78-Å resolution. Two essentially identical molecules of CedV-G were present within the crystallographic asymmetric unit (a.s.u.) (root-mean-square deviation [RMSD] 0.3 Å across 416 equivalent Cα atoms), with the only region of notable structural variation localizing to the β6-S2–S3 loop (Fig 1A). Electron density permitted modelling of the entire receptor-binding domain of CedV-G included in the crystallized construct (residues K209–C622) (Fig 1B).

The receptor-binding domain of CedV-G adopts the canonical six-bladed β-propeller fold used by known HNVs ([18], [26], [39], [40], [41]). Each of the six blades (β1–β6) comprise four antiparallel β-strands that assemble in a toroidal arrangement and form a central depression at the membrane-distal "top" surface of the molecule (Fig 1A). Like the prototypic HNVs, the β2 and β3 blades are

decorated with extended loops that form three short $3_{10}$ helical segments (η1–η3) and three α-helices (α1–α3). Eight disulphide bonds stabilize the fold, seven of which are conserved amongst other HNV-G proteins (Fig S1), indicative of structural importance in stabilizing the β-propeller. The additional disulphide linkage in CedV-G (C310–C376) links blades β2 and β3 at the base of the molecule.

The receptor-binding domain of CedV-G possesses eight N-linked glycosylation sequons, three more than the prototypic HNV-Gs. Electron density corresponding to an asparagine-linked N-acetylglucosamine moiety was observed at seven sites (Figs 1A and B, and S2A). The distribution of glycan sites within CedV-G is consistent with previous studies that implicate the glycan-free β1 and β6 blades in the formation of a homodimeric interface as part of the higher order assembly of virion-displayed HNV-Gs ([40], [42], [43]). Only one of these sequons, $N502^{CedV}$, is conserved with HeV-G ($N481^{HeV}$) and NiV-G ($N481^{NiV}$) at the primary sequence level. Interestingly, glycosylation at $N481^{HeV/NiV}$ has been shown to modulate fusion activation, suggestive that $N502^{CedV}$ may play a similar role ([44]). The otherwise heterogeneous distribution of N-linked glycosylation sequons suggests an absence of functional constraints that would dictate the absolute position of glycan sites across extant HNV-G proteins.

Comparison of CedV-G with unliganded HeV-G (Protein Data Bank [PDB] accession code 2X9M) and NiV-G (PDB: 2VWD) reveals considerable structural similarity within the β-propeller scaffold (Cα atom RMSD of 1.7 Å over 385 residues and 1.6 Å over 371 residues) (Fig 1C), despite substantial primary sequence divergence (Fig S1). Concomitant with genetic proximity, structure-based phylogenetic classification places CedV-G in a cluster of ephrin-tropic HNV-G proteins, closer to the Asiatic prototype viruses HeV-G and NiV-G than the African GhV-G (Fig 1D). Such structure-based phylogenetic classification has demonstrable utility in defining clusters of viral receptor-binding glycoproteins that use common receptors ([18], [45]) and, when applied to CedV-G, supports utilization of ephrins. Whereas core secondary structure elements of the β-propeller scaffold are similar, loop regions exhibit substantial structural differences. Interestingly, structural conservation within the apical loops can be broadly divided into two spatially continuous sections: blades β1–3 are structurally variable and β4–β6 exhibit markedly lower RMSD values (average Cα RMSD of 1.8 and 1.0 Å, respectively) (Fig 1C). ~70% (28/40) of the NiV-G and HeV-G residues that participate in ephrin-B2 recognition are contributed by the structurally conserved β4–β6 blades (Fig S1), suggestive that local structure is constrained by a requirement to maintain ephrin binding.

## CedV-G binds both ephrin-B1 and ephrin-B2

Given the extent and distribution of structural similarities between CedV-G and ephrin-tropic HNVs, we hypothesized that CedV-G likely uses high-affinity ephrin binding to mediate cellular entry. To this end, we examined the binding of soluble ephrins to cell surface expressed HNV-G proteins (Fig 2A). Human embryonic kidney (HEK) 293T cells were transfected with HA-tagged HNV-G glycoproteins (NiV-G, HeV-G, GhV-G, and CedV-G) and titrated against soluble Fc-tagged human B-type ephrins (ephrin-B1–Fc, ephrin-B2–Fc, and ephrin-B3–Fc) to obtain apparent dissociation constants ($K_d$). In

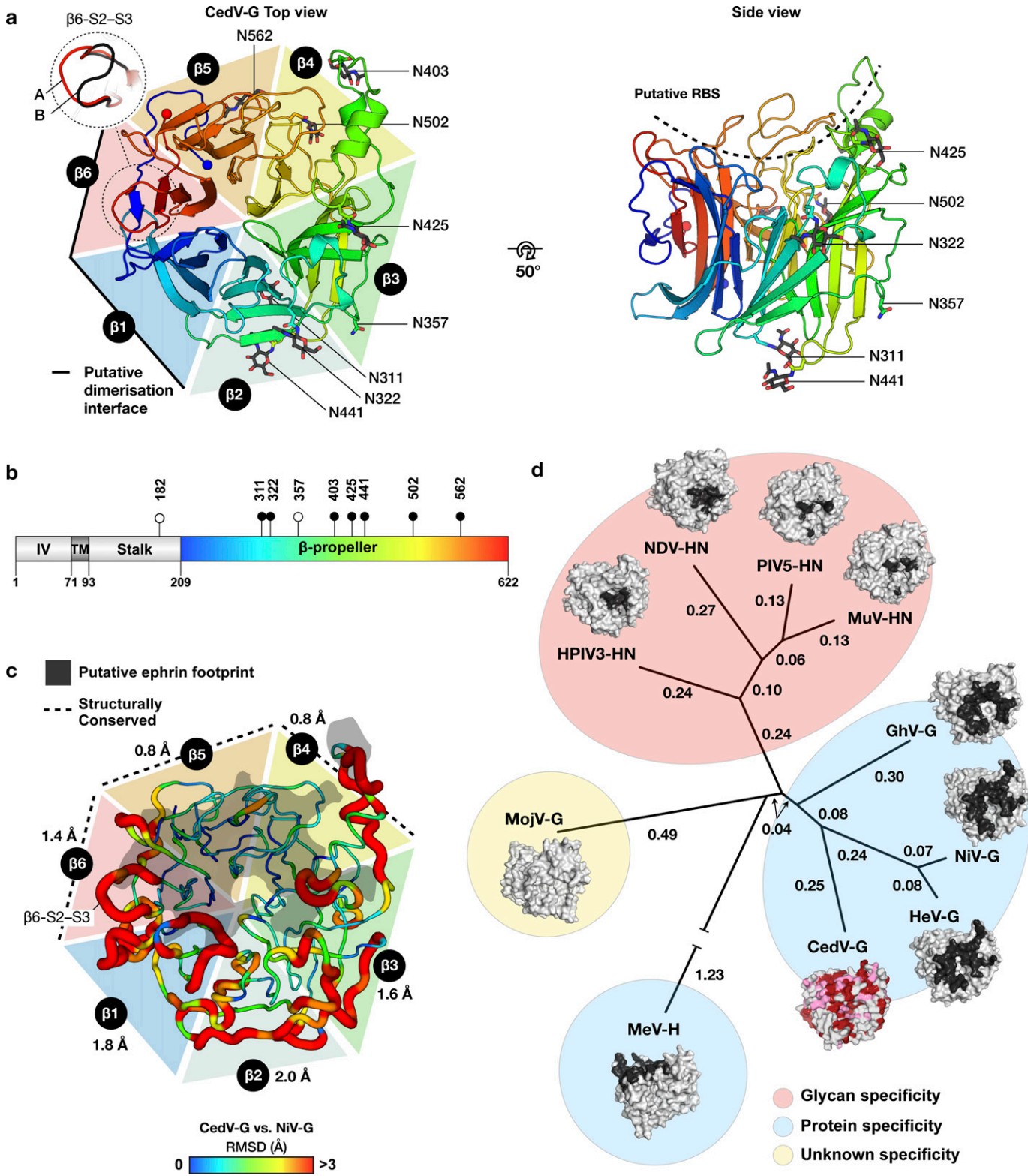

**Figure 1. The CedV-G β-propeller exhibits regions that are structurally conserved with ephrin-tropic HNV-Gs suggestive of a preserved mode of receptor recognition.**
**(A)** The structure of CedV-G is displayed as a cartoon colored from the N- to C terminus (blue to red), with the termini shown as spheres. The approximate extent of each of the six "blades" of the β-propeller is delineated by a colored triangle, labelled β1–β6. The region of highest variability (β6-S2–S3 loop) between the a.s.u. copies (molecule "A" and "B") of the protein is shown with an inset panel (dashed line). Molecule A is shown in rainbow and the β6-S2–S3 loop of molecule B is shown in black. The protein is displayed in a "top" view (left) and rotated 50° to reveal a side view on which the putative receptor-binding site is depicted with a dashed line. N-linked glycans are shown as sticks and colored according to constituent elements. Asparagine residues from all eight N-linked glycosylation sequons are displayed as sticks. The

agreement with previous studies (26, 36, 46, 47), NiV-G bound both ephrin-B2–Fc and ephrin-B3–Fc with nanomolar affinities ($K_d$ = 1.4 and 2.0 nM, respectively), and binding of HeV-G to ephrin-B3–Fc was approximately fivefold weaker than to ephrin-B2–Fc ($K_d$ = 5.3 and 0.86 nM, respectively) (36, 48). Unlike the prototypic HNV-Gs, GhV-G exhibited no detectable interaction with ephrin-B3–Fc but bound strongly to ephrin-B2–Fc ($K_d$ = 0.82 nM) (26). Consistent with our structure-based hypothesis, CedV-G exhibited high affinity binding to ephrin-B2–Fc ($K_d$ = 2.3 nM) but, similar to GhV-G, lacked a titratable interaction with ephrin-B3–Fc. Unexpectedly, we also detected a nanomolar–affinity interaction between CedV-G and ephrin-B1–Fc ($K_d$ = 4.0 nM), a receptor with no precedent of HNV-G binding.

As ephrin-B1 binding was unexpected, we sought to determine whether the high-affinity interactions between CedV-G and both ephrin-B1 and ephrin-B2 have relevance to viral entry. We first tested whether cognate soluble ephrin-B ligands could inhibit the entry of HNV pseudotyped particles (HNVpp) into a mammalian cell type that is permissive to HNV infection, Vero-CCL81 cells. HNV envelope glycoproteins (F and G) were pseudotyped onto a recombinant vesicular stomatitis virus (VSV) expressing a *Renilla* Luciferase (rLuc) reporter gene in place of its endogenous envelope glycoproteins (VSV-ΔG-rLuc). Such VSV-based HNVpp, when used as a surrogate in antibody neutralization assays, have been validated by the Centres for Disease Control and Prevention (CDC; reference 49) as equivalent to the gold standard plaque reduction neutralization titer assay using live HNV (11, 26, 46, 47). We then infected Vero-CCL81 cells with HNVpp in the presence of varying concentrations of soluble ephrin-B–Fc (Fig 2B). To ensure the veracity of our results, we used a fixed quantity of HNVpp pre-determined to give rLuc activity within the linear dynamic range established for this reporter assay (11). Entry of both NiVpp and HeVpp was inhibited by soluble ephrin-B2–Fc and ephrin-B3–Fc but not ephrin-B1–Fc (Fig 2B, top). In contrast, CedVpp entry was inhibited by soluble ephrin-B1–Fc and ephrin-B2–Fc, but not ephrin-B3–Fc, whereas GhVpp was only inhibited by ephrin-B2–Fc (Fig 2B, bottom). These entry inhibition data are consistent with the binding data (Fig 2A) and implicate both ephrin-B1 and ephrin-B2 as receptors for CedV entry.

### Ephrin-B1 and ephrin-B2 support CedV entry

To examine whether ephrin-B1 and ephrin-B2 can function as bona fide receptors for CedV entry, we infected ephrin-negative CHO-pgsA745 cells, engineered to stably express ephrin-B1, -B2, or -B3 (CHO-B1, CHO-B2, and CHO-B3) (46, 47), with CedVpp or NiVpp. Across several logs of viral input, CedVpp was able to infect both CHO-B1 and CHO-B2 cells, but not CHO-B3 cells (Fig 3). Furthermore, NiVpp robustly infected CHO-B2 and CHO-B3 cells but exhibited markedly reduced entry (1,000-fold decrease in relative light unit) into CHO-B1 cells, as has been previously observed (46). Thus, ectopic expression of either ephrin-B1 or ephrin-B2 is sufficient to permit entry of CedVpp into an otherwise non-permissive cell type (CHO-parental) (Fig 3A–C). Furthermore, CedVpp entry into CHO-B2 cells yielded moderate but consistently higher entry levels than CHO-B1 cells (Fig 3C versus 3B), corroborating the affinity-binding (Fig 2A) and ephrin inhibition (Fig 2B) data, indicating that CedV-G uses ephrin-B2 more efficiently than ephrin-B1.

Intriguingly, although both NiV-G and CedV-G exhibit nanomolar affinity binding to ephrin-B2 (Fig 2A), approximately an order of magnitude more CedVpp viral genomes (per mL) were required to achieve entry levels equivalent to NiVpp on CHO-B2 cells (Fig 3C). To eliminate the possibility that NiV glycoproteins are better incorporated into VSV pseudotypes than the CedV equivalents, we performed a western blot analysis, which demonstrated that neither CedV-G nor the fusion-competent cleavage product of CedV-F (CedV-$F_1$) were significantly less incorporated (Fig S3A). Thus, the reduced infectivity of CedVpp is likely a consequence of intrinsically reduced fusogenicity of CedV-F/G compared to NiV-F/G, as evidenced by smaller and fewer syncytia formed by CedV-F/G relative to both NiV-F/G and HeV-F/G, in highly permissive U87 glioblastoma cells (Fig S3B). Despite the lower fusogenicity of CedV-F/G and reduced infectivity, relative to NiVpp on CHO-B2 cells, CedVpp infection was consistently 1–2 logs higher than NiVpp on CHO-B1 cells (Fig 3B). Only at the highest level of viral inoculums (>$10^9$ viral genomes/ml) did NiVpp exhibit very low levels of infectivity on CHO-B1 cells. Altogether, these data indicate that CedV is unique amongst known HNVs in its ability to specifically use both ephrin-B1 and ephrin-B2 for cellular entry.

### Ephrin-B1 supports CedV entry into physiologically relevant primary cells

As endothelial cells constitute the primary targets of HNV infection in vivo (50, 51, 52), we used primary HUVECs to investigate the importance of ephrin-B1 as a functional entry receptor for CedV in a

putative dimerization interface contributed by the β1 and β6 blades is denoted with a solid black line. **(B)** Domain organization and salient features of CedV-G. The attachment-mediating β-propeller domain, transmembrane, and intra-virion regions are labelled. Putative N-linked glycosylation sites are displayed as pins, with sites occupied in the crystal structure colored black. **(A)** The extent of the crystallized construct is colored as a rainbow as in (A). **(C)** Structural comparison of CedV-G and unliganded NiV-G (PDB accession code: 2VWD). Because of the high level of structural similarity between NiV-G and HeV-G (36), an HeV-G comparison is omitted for clarity. RMSD between aligned Cα residues is depicted by both color (in a gradient from blue to red with increased RMSD) and tube width (thin to thick with increased RMSD). Residues that failed to align or exhibited RMSDs greater than 3 E were assigned values of 3 E. The average RMSD across each blade is displayed next to the respective blade label and the more structurally conserved region of the molecule (β4–β6) is indicated with a dashed line. The NiV-G–ephrin-B2 interface is displayed as a grey shadow superposed onto the structure of CedV-G. **(D)** Structure-based phylogenetic analysis of paramyxovirus receptor-binding proteins places CedV-G amongst ephrin-using HNV-G proteins. Pairwise distance matrices were calculated with Structural Homology Program (37) and plotted with PHYLIP (38), using the structures of unliganded receptor-binding glycoproteins, where available. The corresponding structures are shown in surface representation with previously characterized receptor-binding surfaces shown in dark grey. The structure of CedV-G is colored according to sequence conservation with NiV-G, identical residues are red and similar residues are pink. Measles virus hemagglutinin (MeV-H) branch is truncated for illustrative purposes. Structures used for the analysis were the Ghanaian bat henipavirus G (GhV-G; PDB 4UF7), Nipah virus G (NiV-G; PDB 2VWD), Hendra virus G (HeV-G; PDB 2X9M), CedV-G, MeV-H (PDB 2RKC), Mòjiāng virus G (MojV-G; PDB 5NOP), human parainfluenza virus 3 hemagglutinin-neuraminidase (HPIV3-HN; PDB 1V3B), Newcastle disease virus HN (PDB 1E8T), parainfluenza virus 5 HN (PIV5-HN; PDB 4JF7), and mumps virus HN (MuV-HN; PDB 5B2C). Branches of the resultant tree are labelled with the calculated evolutionary distances.

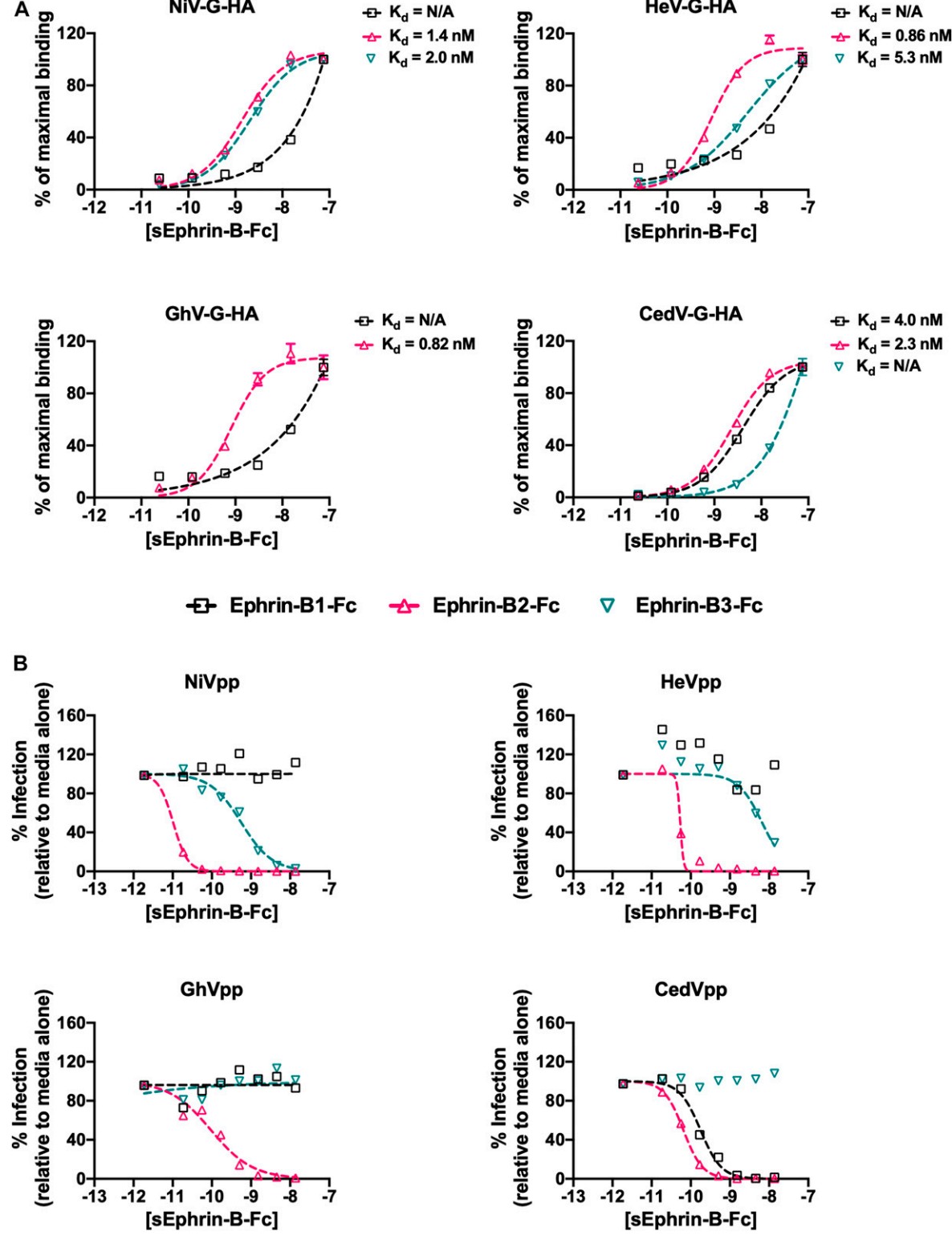

**Figure 2. Ephrin-B1 and ephrin-B2 bind CedV-G.**

**(A)** Increasing concentrations of soluble ephrin-B–Fc ($10^{-11}$ to $10^{-7}$ M) were added to HNV-G transfected HEK-293T cells and binding (measured as GMFI values) was assessed by flow cytometry using Alexa Fluor 647–labelled anti-human Fc antibodies. Four-parameter dose–response or logistic (4PL) curves were generated by nonlinear regression using GraphPad Prism. Data from GMFI values are displayed as percent of maximal binding, with the maximal binding value at the highest concentration of ligand used set to 100%. The bottom of each 4PL curve was constrained to have a constant value of zero. The reported $K_d$ (dissociation constant) corresponds to the ephrin ligand concentration [sEphrin-B3–Fc], at which 50% maximal binding is achieved. A value of N/A refers to data that could not be fitted

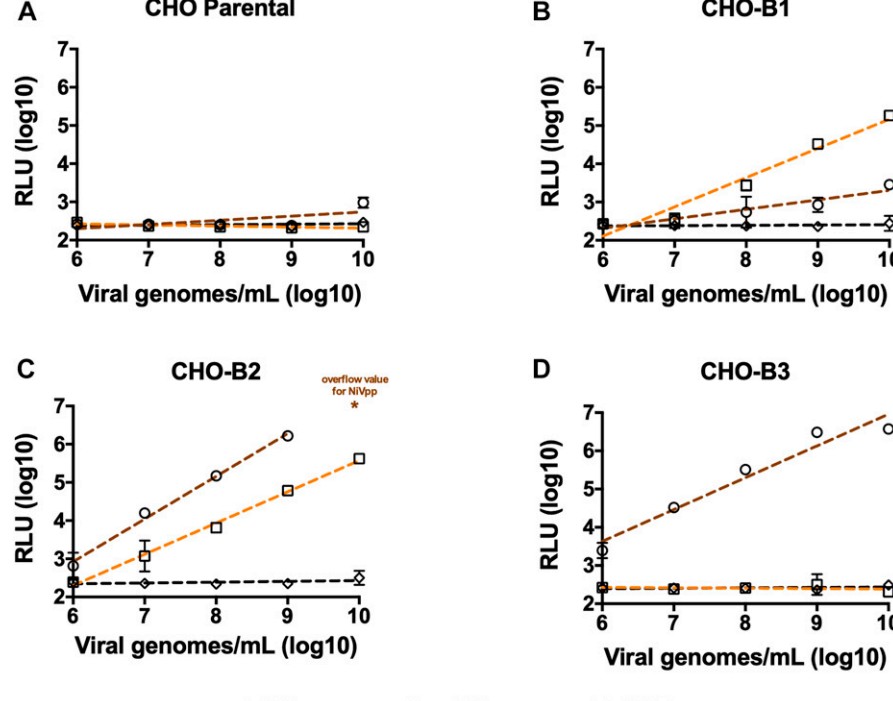

**Figure 3. Ectopic expression of ephrin-B1 or ephrin-B2 is sufficient to confer CedVpp entry into a non-susceptible cell type.**
**(A, B, C, D)** NiVpp, CedVpp, and BALDpp (VSV pseudotypes bearing no viral glycoprotein) were used to infect (A) CHO-pgsA745 cells (a naturally ephrin-negative cell line) or CHO-pgsA745 cells that stably express (B, C, D) ephrin-B1 (CHO-B1), ephrin-B2 (CHO-B2), or ephrin-B3 (CHO-B3), respectively, over a range of viral inoculum (viral genomes/mL). Entry was measured as described in the legend to Fig 2. The asterisk indicates an RLU value above the maximum limit of detection. RLU were plotted against numbers of viral genome copies per milliliter and fitted to a linear regression (dashed lines) using GraphPad Prism. Data shown are the averages of three independent biological replicates ± SE.

physiologically relevant cell type. Real-time quantitative PCR (qPCR) analysis of B-class ephrin transcripts within primary HUVECs revealed the presence of mRNA encoding both ephrin-B1 and ephrin-B2, whereas ephrin-B3 mRNA was undetectable (Fig 4A). After confirmation of active transcription of ephrin-B1 and ephrin-B2 in HUVECs, soluble envelope competition assays were performed to assess HNVpp entry (Fig 4B and C).

Preincubation of HUVECs with saturating quantities of sNiV-G–Fc completely inhibited NiVpp entry, suggesting that all available cell surface displayed ephrin-B2 molecules were sequestered by soluble NiV-G and thus viral entry was not supported in the absence of an alternate cognate receptor for NiV, namely, ephrin-B3. Conversely, under identical conditions, CedVpp entry was still supported at ~50%, evidencing ephrin-B1-mediated entry of CedVpp in the absence of available ephrin-B2 (Fig 4B). Entry of both NiVpp and CedVpp was completely abrogated by preincubation with soluble Eph-B3 receptor (sEph-B3–Fc), an Eph receptor that binds to all three B-class ephrin ligands with similar affinities (53), further confirming that viral entry is ephrin-dependent (Fig 4C). These data reveal that endogenously expressed ephrin-B1 supports CedVpp entry in a physiologically relevant cell type, validating the role of ephrin-B1 as a functional entry receptor for CedV.

## The CedV-G–ephrin-B1 structure reveals a conserved HNV-G–ephrin interaction mode

After our findings that CedV was able to use a previously unreported receptor repertoire, we sought to delineate the molecular features of ephrin binding and ligand selectivity. To this end, we solved the crystal structure of the attachment-mediating β-propeller of CedV-G in complex with the extracellular β-barrel domain of human ephrin-B1, to 4.07-Å resolution. Five complexes populated the crystallographic a.s.u. and displayed no appreciable structural variation, within the limits of the resolution. Analyses herein concern the complex comprising chains "B" and "D," which was selected for superior quality electron density and the model quality permitted as a consequence (Fig S4).

Comparison with the unliganded structure of CedV-G reveals little overall structural variation (0.4 Å RMSD over 417 equivalent Cα) within the ephrin-B1–bound CedV-G scaffold. Notably, the β6-S2–S3 loop, which is conformationally distinct in each of the unliganded CedV-G crystallographic a.s.u. copies (Fig 1A), is also the region of greatest structural variability when comparing the ephrin-B1 bound and unbound states of CedV-G. Structural plasticity within this region is observed in other HNV-G proteins and their ephrin-bound complexes (39, 40).

unambiguously to a 4PL curve, that is, binding of soluble ephrin was not titratable or saturation could not be achieved even at concentrations up to 100 nM. Values for ephrin-B3–Fc binding to GhV-G are not displayed, as no detectable binding over background was observed. Data shown are the averages of three independent biological replicates ± SE. **(B)** NiV-F/G (NiVpp), HeV-F/G (HeVpp), GhV-F/G (GhVpp), and CedV-F/G (CedVpp) VSV-ΔG-rLuc pseudotyped viruses were used to infect Vero CCL81 cells in the presence of increasing amounts of soluble ephrin-B1–Fc, ephrin-B2–Fc, and ephrin-B3–Fc fusion proteins ($10^{-12}$ to $10^{-8}$ M). 4PL curves were generated using GraphPad Prism as above. Data from relative light unit(s) (RLU) values are displayed as percent of maximal infection, defined as the RLU achieved in the presence of media alone, which is set to 100%. The top and bottom of each 4PL curve was constrained to have a constant value of 100 and 0, respectively. Data shown are the averages of three independent biological replicates ± SE.

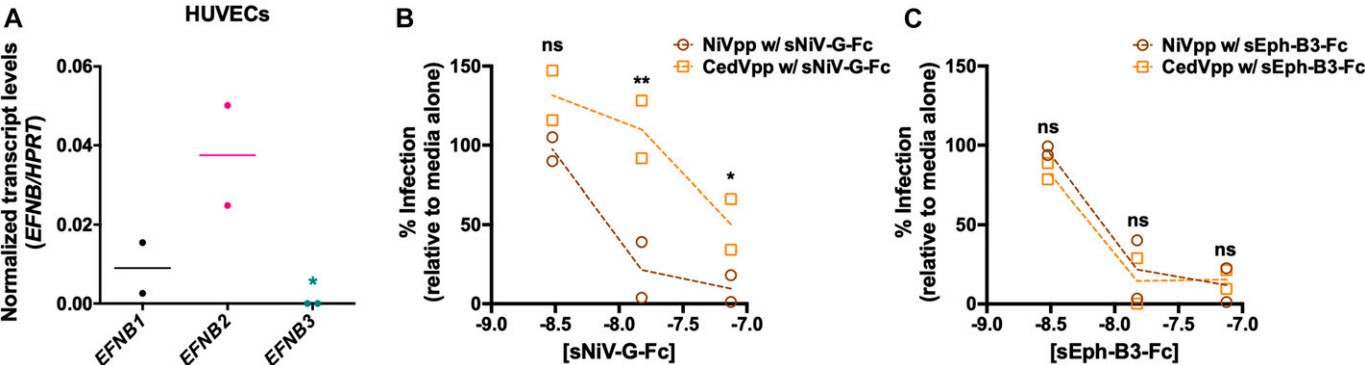

**Figure 4. Ephrin-B1 facilitates CedVpp entry into biologically relevant primary HUVECs.**
**(A)** Active transcription of ephrin-B1, ephrin-B2, and ephrin-B3 in primary HUVECs was determined by qPCR. Transcript levels are shown normalized to hypoxanthine phosphoribosyltransferase (HPRT) transcripts. Ephrin-B3 transcript levels were below the limit of detection (asterisk). Data shown are the individual data points from two independent biological replicates. Horizontal dashes represent the mean from the two replicates. **(B, C)** NiVpp, CedVpp, and VSVpp were used to infect primary HUVECs in the presence of increasing amounts ($10^{-9}$ to $10^{-7}$ M) of Fc-tagged (B) soluble NiV-G (sNiV-G–Fc) or (C) soluble Eph-B3 receptor (sEph-B3–Fc). Entry was measured as in Fig 2. Data are shown as percent infection relative to the signal achieved when the viruses are incubated in the presence of media alone. Data shown are the individual data points from two independent biological replicates. Dashed lines connect the means from the duplicate data. Statistical significance for this entry inhibition assay was tested with a two-way ANOVA with Holm-Sidak's correction for multiple comparisons, n/s denotes no significance, * denotes $P < 0.05$, ** denotes $P < 0.005$.

CedV-G engages ephrin-B1 with an overall binding mode that is similar to that used by other HNV-G proteins, when recognizing their respective ephrin receptors (Fig 5A–C) (26, 36, 48). CedV-G and ephrin-B1 form a 1:1 complex with an extensive molecular interface that buries a combined surface area of 2,900 $\text{Å}^2$ (1,450 $\text{Å}^2$ per component). The buried interface is larger than that previously characterized for other HNV-G–ephrin complexes (NiV-G–ephrin-B2 = 2,800 $\text{Å}^2$; NiV-G–ephrin-B3 = 2,700 $\text{Å}^2$; HeV-G–ephrin-B2 = 2,600 $\text{Å}^2$; GhV-G–ephrin-B2 = 2,300 $\text{Å}^2$, calculated using the PDBePISA server (54)). Ephrin-B1 is bound at the top of the molecule and forms an interface that chiefly comprises residues from the structurally conserved β4–6 blades of CedV-G, which contribute ~70% (~2,000 $\text{Å}^2$) of the total buried surface area (Figs 1, 5C, and S1). Despite localizing to the interface, the glycan at N425 extends away from ephrin-B1 and does not appear to participate in receptor binding.

Like the other HNV-G–ephrin complexes, the G–H loop of ephrin-B1 (residues 113–129$^{ephrin-B1}$) constitutes a major interacting region and is inserted into the central depression on top of CedV-G, contributing 1,700-$\text{Å}^2$ buried surface area to the molecular interface (Fig 5A–C). Interestingly, despite sequence variation and pronounced structural dissimilarity in their unliganded states (55), the G–H loops of ephrin-B1 and ephrin-B2 adopt a strikingly similar conformation in their HNV-G–bound states (Fig 5C). Indeed, the Greek-key fold of the bound ephrin-B1 ectodomain is highly similar to that of NiV-G–bound ephrin-B2 (1.0-Å RMSD across 135 aligned Cα atoms), although less similar to ephrin-B3 (1.6 Å RMSD across 126 aligned Cα atoms). Differences between the structural states of both CedV-G and ephrin-B1 may represent an induced-fit mechanism of ephrin recognition, which has been postulated for other ephrin-tropic HNVs (36, 39, 40), or selection from a conformational equilibrium.

### Ephrin-B1 and ephrin-B2 are bound by the same site on CedV-G

Given the striking structural similarities between CedV-G–ephrin-B1 and other HNV-G–ephrin complexes (Fig 5B), we hypothesized that CedV-G likely uses the same receptor-binding site to engage ephrin-B2.

To assess this, we determined pseudotyped virus entry into CHO-B1 and CHO-B2 cells in the presence of competing soluble B-class ephrin ligands. As expected (46, 56, 57), ephrin-B2–Fc and ephrin-B3–Fc inhibited NiVpp entry into CHO-B2 cells, whereas ephrin-B1–Fc failed to strongly inhibit entry at concentrations as high as 10 nM (Fig 5D, middle panel), confirming that ephrin-B2 and ephrin-B3 are each bound by the same site on NiV-G (36, 48). Similarly, both ephrin-B2–Fc and ephrin-B1–Fc inhibited CedVpp entry into CHO-B2 cells (Fig 5D, right panel), evidencing the ability of ephrin-B1 to block ephrin-B2–dependent CedVpp entry through competition for an overlapping binding site on CedV-G. Moreover, ephrin-B2–Fc inhibited CedVpp entry into CHO-B1 cells (Fig 5D, left panel). In both CHO-B2 and CHO-B1 cells, ephrin-B2–Fc–mediated inhibition of CedV-G was more potent than ephrin-B1–Fc (Fig 5D), further supporting our binding (Fig 2) and entry (Fig 3) data that suggest ephrin-B2 is more efficiently used than ephrin-B1.

To further validate our hypothesis, we performed soluble envelope competition assays in which CHO-B2 cells were incubated with soluble NiV-G–Fc, before infection with NiVpp or CedVpp (Fig 5E, middle and right panels, respectively). Entry of NiVpp and CedVpp on CHO-B2 cells was completely inhibited by NiV-G–Fc, indicative that CedV-G and NiV-G recognize a common interface on ephrin-B2, which, in conjunction with the structures of CedV-G (Fig 1) and CedV-G–ephrin-B1 (Fig 5A–C), supports a universal mode of ephrin recognition across ephrin-tropic HNVs. Importantly, CedVpp entry into CHO-B1 cells was unaffected by saturating amounts of NiV-G–Fc (Fig 5E, left panel). Thus, despite a shared mode of receptor engagement across ephrin-tropic HNVs, these data indicate that CedV-G possesses subtle but distinct features within its receptor-binding site that determine utilization of its idiosyncratic receptor repertoire.

### Accommodation of the YM motif of ephrin-B1 is a key determinant of receptor specificity

To determine the molecular features that underscore the distinct ephrin specificity of CedV-G, we examined both the structural and

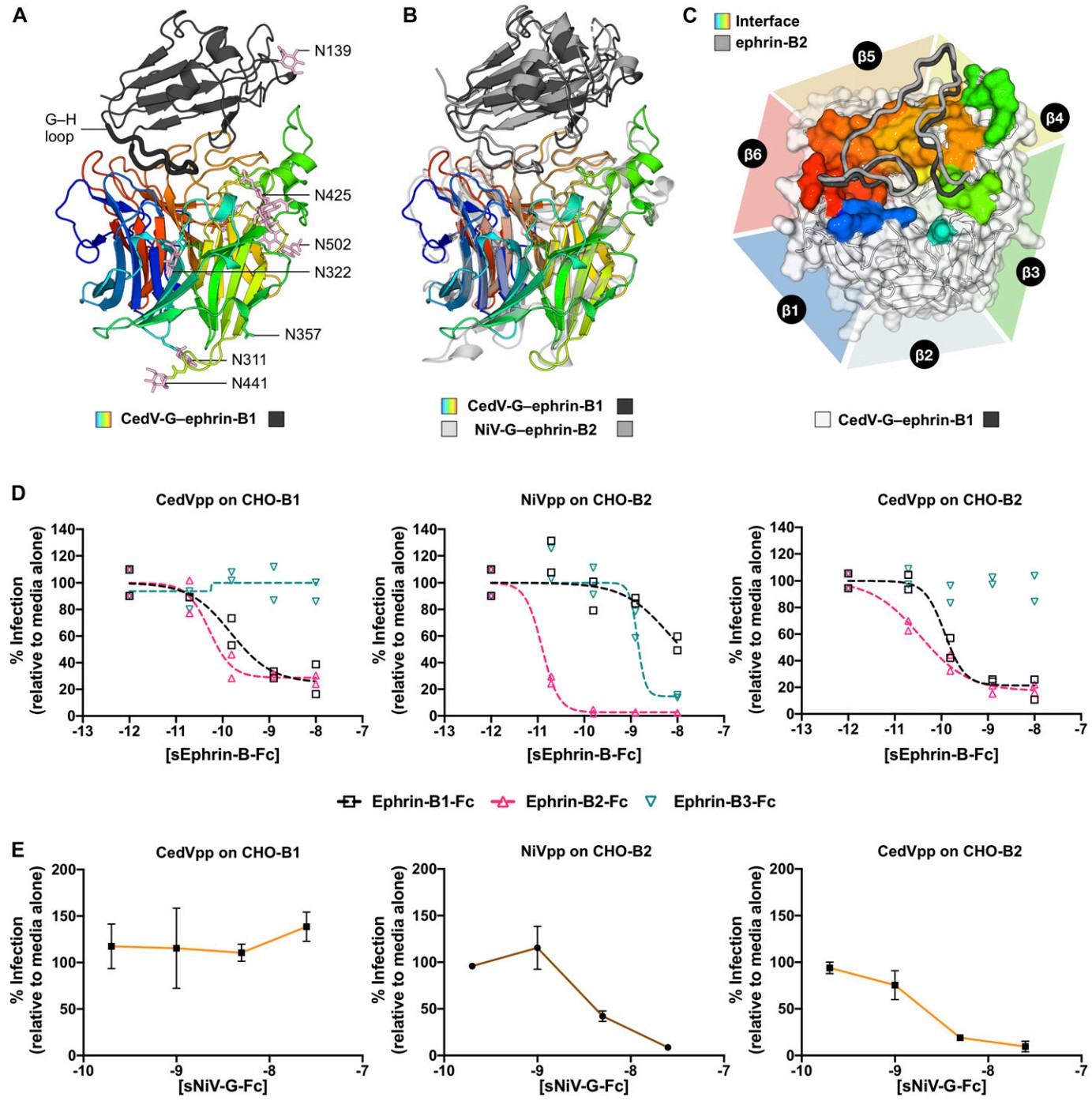

**Figure 5. CedV-G binds ephrin-B1 and ephrin-B2 at a conserved and overlapping binding site.**
**(A)** The crystal structure of CedV-G in complex with ephrin-B1 reveals a conserved mode of receptor engagement across ephrin-tropic HNV-G proteins. The receptor-binding domain of CedV-G (colored in a gradient from blue to red, from N- to C terminus) forms a 1:1 complex with the extracellular domain of ephrin-B1 (dark grey). The principal interaction region of ephrin-B1, the "G–H loop," is displayed as a thick tube for clarity. Modelled N-linked glycan moieties (pink) and the asparagine residues of all putative N-linked glycosylation sequons are shown as sticks. **(B)** Superposition of NiV-G–ephrin-B2 (PDB: 2VSM) on CedV-G–ephrin-B1. The NiV-G–ephrin-B2 complex is colored in light grey, with NiV-G shown as transparent for clarity. **(C)** Comparison of bound ephrin-B1 and ephrin-B2 molecules. CedV-G is shown as a white transparent surface with the ephrin-B1 interface colored according to sequence position as in panel (A). Regions of ephrin-B1 (dark grey) and ephrin-B2 (light grey) bound by CedV-G and NiV-G, respectively, are shown as cartoon tubes. β-propeller blades are delineated with triangles that are colored according to sequence position as in panel (A). **(D)** Ephrin-B1 and ephrin-B2 compete for binding to CedV-G. NiVpp and/or CedVpp were used to infect CHO-B2 cells (*middle and right panels*) or CHO-B1 cells (*left panel*), as indicated, in the presence of increasing amounts of soluble ephrin-B1–, ephrin-B2–, and ephrin-B3–Fc ($10^{-12}$ to $10^{-8}$ M). Entry was measured as in Fig 2. 4PL dose–response curves were generated as in Fig 2 and based on values displayed as percentages of infection with the RLU achieved in the presence of media alone set to 100%. The top of each fit curve was constrained to a have a constant value of 100. Data shown are the individual data points from two independent biological replicates with the fit curves shown in dashed lines. **(E)** NiV-G and CedV-G compete for binding to ephrin-B2.

functional implications of two hydrophobic motifs that differ between ephrin-B1 and ephrin-B2. Although the G–H loops of B-type ephrin paralogs are highly conserved (Fig 6A), ephrin-B1 and ephrin-B2 differ by the presence of a $Tyr^{121}$–$Met^{122}$ (YM) or $Leu^{121}$–$Trp^{122}$ (LW) motif, which have previously been shown to be critical determinants of differential receptor utilization (46). Inspection of the CedV-G–ephrin-B1 complex structure reveals that the side chain of $Tyr^{121-ephrin-B1}$ is inserted into a large hydrophobic cavity that is unique to CedV-G (Fig 6B and C). Indeed, the equivalent region within NiV-G ($Trp^{504}$), HeV-G ($Trp^{504}$), and GhV-G ($Trp^{514}$) is, in all instances, occluded by a tryptophan side chain that protrudes toward the toroidal axis of the β-propeller, rather than being sequestered in the opposing direction as is the equivalent residue, $Tyr^{525-CedV-G}$ (Fig 6C). Thus, it is likely that the expanded hydrophobic cavity of CedV-G is critical to ephrin-B1 recognition as it circumvents steric clashes that would preclude its recognition by other HNV-Gs.

To interrogate this structure-based hypothesis, we tested pseudotyped viral entry into CHO-pgsA745 cells stably expressing a panel of wild-type and mutant ephrin ligands (Fig 6D). These stable cell lines had been previously demonstrated to express similar levels of the indicated wild-type and mutant ephrin ligands, as measured by sEph-B3–Fc binding (46). As anticipated, NiVpp and CedVpp were able to enter cells expressing ephrin-B2$_{WT}$, whereas ephrin-B1$_{WT}$ supported only CedVpp entry. Replacement of the ephrin-B2 LW motif with the equivalent YM residues of ephrin-B1 (ephrin-B2$_{YM}$) severely impaired the ability of ephrin-B2$_{YM}$ to support NiVpp entry, although did not prevent it entirely. Conversely, the reciprocal exchange of the YM to LW motif in ephrin-B1 (ephrin-B1$_{LW}$) rendered it capable of supporting some level of NiV entry. Intriguingly, ephrin-B1$_{LW}$–mediated entry of CedVpp was increased relative to ephrin-B1$_{WT}$, suggestive that the YM motif is less efficiently used than LW for CedV entry, in the context of an ephrin-B1 background. By contrast, introduction of the YM motif into ephrin-B2 induced no appreciable difference in its ability to support CedVpp entry. The differential response to the YM/LW motifs, when presented in the context of distinct ephrin backgrounds, highlights the importance of the myriad surrounding interactions that comprise the extensive protein–protein interface. Taken together, this integrated structural and functional analysis suggests that the ability of CedV-G to effectively accommodate the YM motif of ephrin-B1, in a sterically unrestrained cavity, is a critical determinant that directs ephrin-B1 utilization.

## Discussion

Viral surveillance continues to revise the known geographic coverage and biological diversity of HNV species (12). Although the identification of HeV and NiV was a consequence of their emergence as the etiological agents of severe respiratory and neurological disease, targeted virus discovery efforts have demonstrated utility in identifying novel HNVs yet to be linked with symptomatic

illness (10, 11, 19). One such virus, CedV, represents only the third HNV species to be isolated and remains the sole member of the genus confirmed as nonpathogenic (19). Combined, the starkly contrasting virulence and the ecological and biological similarities between CedV and the deadly HNVs demarcates CedV as a valuable model with which to assess HNV functional diversity and the determinants of pathogenesis.

Through adoption of a combined structural and functional approach, we sought to determine the receptor repertoire used by CedV-G and uncover molecular features that dictate its specificity. We demonstrate that CedV-G uses a structurally constrained receptor-binding architecture to mediate recognition of highly conserved ephrin receptors using an analysis that relates local structural conservation to discrete regions that are functionally constrained amongst genetically diverse HNVs (Fig 1C). By demonstrating ephrin-B2–mediated cellular entry of CedV, we confirm previous studies (19, 35) and provide evidence for a conserved receptor recognition mode (Fig 5). Unexpectedly, we detected high affinity binding to ephrin-B1, a B-class ephrin with no precedent of supporting HNV entry (Figs 2 and 3) and demonstrate that the interaction is sufficient to permit viral entry into pathobiologically relevant primary endothelial cells (Fig 4). Furthermore, we present crystal structures of CedV-G, in both unbound and ephrin-B1–bound states (Figs 1 and 5), which reveal a common mode of receptor recognition across distinct B-type ephrin ligands and diverse HNV-G proteins.

The bipartite nature of structural conservation within CedV-G (Fig 1C) is indicative of contrasting selective pressures which, in blades β1–β3 permit diversification, but in the β4–β6 blades, constrain variation to maintain high-affinity ephrin binding. Like NiV and HeV, CedV uses ephrin-B2 more efficiently than its cognate alternate receptor, ephrin-B1 (or ephrin-B3 for NiV and HeV) (Figs 2 and 3) (36, 46). Furthermore, despite its idiosyncratic receptor repertoire, structure-based phylogenetic analysis co-localizes CedV-G with other ephrin-tropic HNV-Gs, all of which are unified in using ephrin-B2 for cellular entry (Fig 1D). Combined, the more effective ephrin-B2 usage and structure-based phylogenetic association of all ephrin-tropic HNV-Gs implicate ephrin-B2 as a receptor used by ancestral HNVs, alongside which coincident alternate receptor specificities may have arisen.

Despite evident commonalities of HNV-G–mediated ephrin recognition, the structure of CedV-G–ephrin-B1 reveals unique molecular properties that provide a structure-based rationale for receptor specificity. Modulation of the side chain orientation of a single amino acid in CedV-G ($Y^{525}$) relieves a steric barrier that likely precludes ephrin-B1 recognition by other HNV-Gs. Given that only minor changes to ephrin-B1 are required for it to support NiV entry (Fig 6D) (46), and the existence of relatively subtle structural differences within HNV-G receptor-binding sites, it is possible that the existence or acquisition of ephrin-B1 tropism in extant HNVs may not be uncommon. Although the molecular features that preclude ephrin-B3 utilization by CedV-G remain unclear, our structural hypothesis

NiVpp or CedVpp were used to infect CHO-B2 *(middle and right panels)* or CHO-B1 *(left panel)* cells in the presence of increasing amounts of soluble Fc-tagged NiV-G (sNiV-G–Fc) ($10^{-10}$ to $10^{-7}$ M). Data are shown as percent infection relative to the signal achieved when the viruses were incubated in the presence of media alone. Data shown are the averages of three independent biological replicates ± SE.

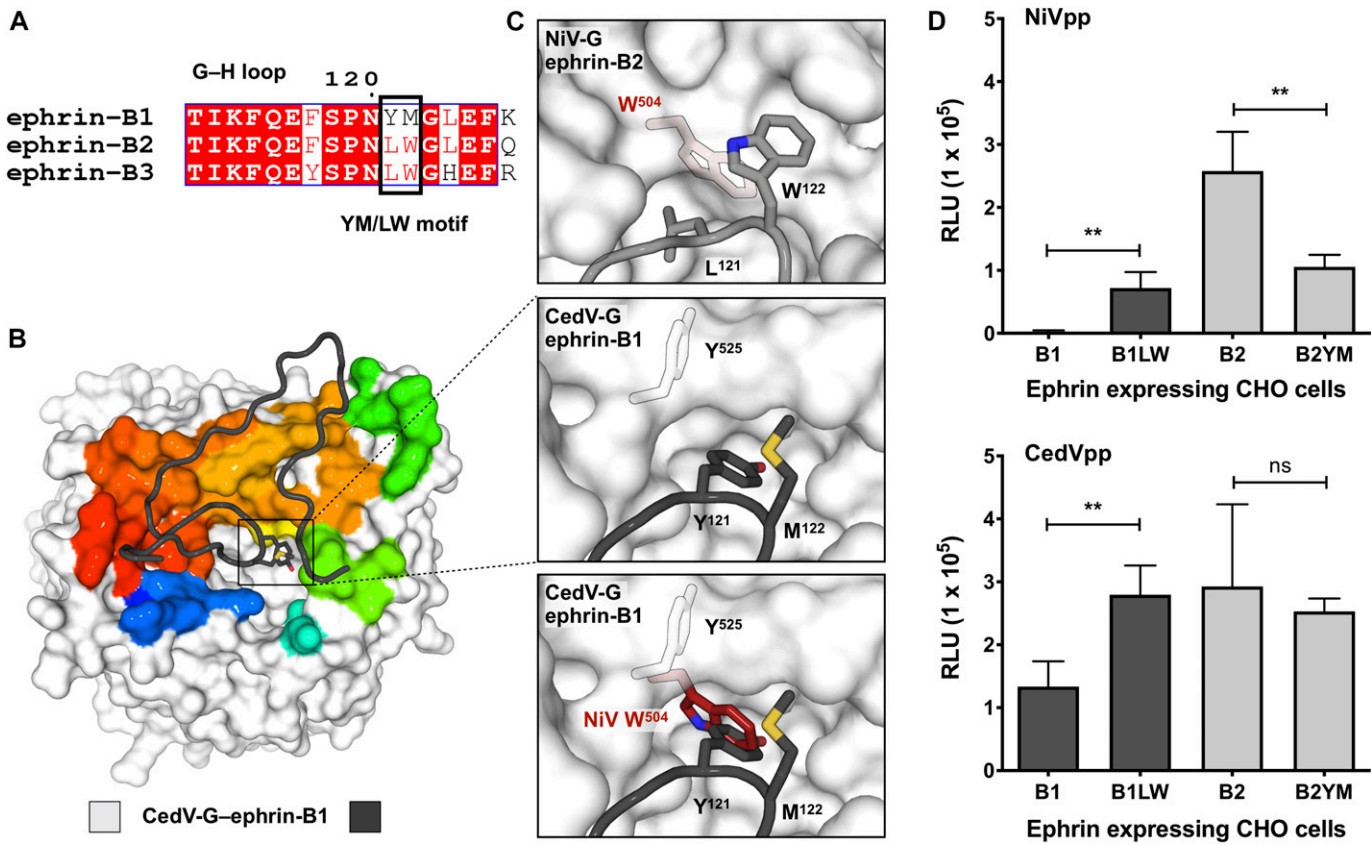

**Figure 6. Accommodating the YM motif is critical for ephrin-B1 utilization by CedV.**
**(A)** Sequence alignment of the G–H loop region of human B-type ephrins. Absolutely conserved residues are highlighted red, partially conserved residues are colored red, and non-conserved residues are black. Residues that constitute the YM/LW motif are outlined with a black box. Sequences are numbered according to ephrin-B1. Alignments were determined by MultAlin (80) and plotted using ESPript (81). **(B)** Structure of CedV-G–ephrin-B1. CedV-G is displayed as a white surface with the ephrin-B1 interface colored according to the sequence position (blue to red, N- to C terminus). The principal interacting region of ephrin-B1 is shown as a dark grey cartoon tube, with the side chains of Y121 and M122, the "YM motif," shown as sticks and colored according to constituent atoms. **(C)** Detailed view of the YM motif of ephrin-B1 (boxed region in a; *middle*) and the equivalent view of the LW motif of NiV-G–bound ephrin-B2 (light grey; *top*). (*bottom*) Overlay of the NiV-G residue, W504 (red) onto CedV-G–ephrin-B1, demonstrates potential steric overlap between NiV-G and ephrin-B1. The side chain of Y525 in CedV-G is sequestered away from the receptor-binding site. Side chains of key residues are shown as sticks and colored according to constituent atoms. **(D)** CedV is more tolerant to substitution of the YM/LW motif than NiV. CHO cells expressing both wild-type ephrins (B1 and B2) and mutants with reciprocally exchanged LW/YM motifs (B1LW and B2YM) were infected with NiVpp (*top*) or CedVpp (*bottom*). Entry was assessed and quantified as in Fig 2. Data represent the average of quadruplicate measurements ± SE. Statistical significance for the indicated comparisons were evaluated using a two-tailed unpaired *t* test, ** denotes *P* < 0.005 and n/s denotes no significance.

suggests that acquired ephrin-B1 specificity does not necessarily come at the expense of ephrin-B3 usage, as the LW motif is common to both ephrin-B2 and ephrin-B3.

The acquisition of alternate receptor specificities, and the species and cell type tropism thereby engendered, is of acute biomedical and agricultural importance (27). For example, the systemic dissemination and multi-organ vasculitis associated with NiV and HeV is consistent with expression of ephrin-B2 in endothelial cells (56, 58) and peripheral organs, such as the kidney, lung, and spleen, wherein ephrin-B3 is less abundant. However, central nervous system disease pathologies (59, 60), which are the ultimate cause of death in fatal NiV and HeV infection, are likely a consequence of markedly increased and specific expression of ephrin-B3 in multiple brain regions (Fig S5) (61, 62, 63).

Although the absence of IFN antagonism in CedV likely plays a critical role in determining its apathogenic phenotype (19, 20), an inability to use ephrin-B3 may also be a contributing factor. Interestingly,

although ephrin-B1 is expressed at low to negligible levels in the central nervous system, its expression in other tissues is both more widespread and often greater in magnitude than ephrin-B2. Of note is the relatively high expression of ephrin-B1 in the lung, esophagus, and salivary glands (Fig S5), which suggests that ephrin-B1 utilization could augment aspects of oropharyngeal transmission postulated for HNVs (64), especially because ephrin-B1 is almost as conserved as ephrin-B2 across mammalian species (96–99% sequence similarity). Thus, although the pathobiological and ecological implications of ephrin-B1 tropism are presently unclear, our study sets a precedent for ephrin-B1 utilization and in so doing expands the known repertoire of HNV cellular entry receptors used by this group of lethal human pathogens. Whilst the preprint version of this article was under peer-review (65 Preprint), Laing et al reported similar results (66).

Finally, in light of our structure–function analyses, it is plausible that the barrier for ephrin-B1 usage may not be high, and that other

HNVs may acquire, or have already acquired, an expanded repertoire of B-class ephrin receptors that could modulate pathogenicity and transmissibility. Characterization of receptor tropism characteristics should be central to future surveillance efforts that aim to identify and assess new HNVs, to elucidate the association between pathogenicity and expanded receptor usage.

# Materials and Methods

### Protein production

Independent expression vectors for the putative six-bladed $\beta$-propellerdomain of CedV-G (residues 209–622; National Center for Biotechnology Information [NCBI] reference sequence: YP_009094086.1) and the G-interacting N-terminal extracellular domain of ephrin-B1 (residues 29–167; NCBI reference sequence NP_004420.1) were generated by PCR amplification from codon-optimized synthetic cDNA templates (GeneArt; Life Technologies) and subsequent restriction-based cloning into the pHLsec mammalian expression vector (67).

HEK-293T cells (American Type Culture Collection CRL-1573) were transiently transfected with the desired protein constructs and expressed in the presence of the class 1 $\alpha$-mannosidase inhibitor, kifunensine (68). For the generation of CedV-G–ephrin-B1 complexes, cells were co-transfected with a 1:1.5 mass ratio of CedV-G to ephrin-B1 cDNA. Cell supernatants were harvested and clarified 72 h posttransfection, after which supernatants were diafiltrated against a buffer containing 10 mM Tris (pH 8.0) and 150 mM NaCl (ÄKTA Flux diafiltration system; GE Healthcare). In all instances, glycoproteins and their complexes were purified by a tandem immobilized nickel affinity and size exclusion chromatography strategy using HisTrap HP and Superdex increase 200 10/300 GL columns (GE Healthcare), equilibrated in 10 mM Tris (pH 8.0) and 150 mM NaCl, respectively. To aid crystallogenesis of unliganded CedV-G, high-mannose–type N-linked glycans were trimmed by partial enzymatic deglycosylation with endoglycosidase F1 (25°C for 18 h).

For the production of Fc-tagged HNV-G proteins used in cellular assays, codon-optimized DNA fragments corresponding to the receptor-binding domains of NiV-G (residues 183–602; NCBI reference sequence: NC_002728.1) and CedV-G (residues 209–622; NCBI reference sequence: YP_009094086.1) were cloned into the pHL-FcHis vector (67). Transfection and protein purification procedures were identical to those used for the production of non-Fc–tagged proteins, detailed above. In all instances, kifunensine was not present during the expression of HNV-G–Fc.

### Crystallization and structure determination

Endoglycosidase F1–treated CedV-G was concentrated to 6.7 mg/ml and subjected to room temperature crystallization screening using the sitting-drop vapor diffusion method (69). A single crystal of CedV-G was observed after 15 d in a condition comprising 5% (wt/vol) polyethylene glycol 6000 and 0.1 M MES pH 6.0. In addition, CedV-G–ephrin-B1 containing high-mannose glycans derived from expression in the presence of kifunensine was concentrated to 12.2 mg/ml. Crystals formed after 132 d in a condition comprising 25%

polyethylene glycol 1500 (wt/vol), 0.1 M PCB (sodium propionate, sodium cacodylate, and BIS-TRIS propane) pH 5.0 and 0.02 M cadmium chloride dihydrate (70).

Crystals were harvested and cryoprotected by transfer into a drop of the respective mother liquor supplemented with 25% glycerol (vol/vol) before flash cooling in liquid nitrogen. X-ray diffraction data were collected for CedV-G (to 2.78-Å resolution) and CedV-G–ephrin-B1 (to 4.07-Å resolution) at beamlines I02 and I24 at Diamond Light Source (UK), respectively. Crystal data were indexed, integrated, and scaled using XIA2 (71). The structure of CedV-G was solved by molecular replacement, implemented in PHASER (72), using the structure of unliganded NiV-G (PDB accession code 2VWD) as a search model. The structures of CedV-G (reported here) and ephrin-B2 (derived from the NiV-G–ephrin-B2 complex [PDB: 2VSM]), were used as independent search models to solve the structure of CedV-G–ephrin-B1. In both instances, iterative rounds of model building and refinement were performed using COOT (73), PHENIX (74), and REFMAC (75), using non-crystallographic symmetry restraints and translation/libration/screw parameterization. The structure of CedV-G–ephrin-B1 was refined using grouped B-factors and reference model restraints derived from the higher resolution CedV-G structure. Conformational validation of N-linked glycans was performed using Privateer (76). Data collection and refinement statistics are presented in Table S1.

### Structure-based phylogenetic analysis

Available structures of unique paramyxovirus receptor-binding glycoproteins (G/H/HN) were used to construct a structure-based phylogenetic tree. Structures used were as follows: human parainfluenza virus 3, HPIV3-HN (PDB accession code: 1V3B) (77); Newcastle disease virus, NDV-HN (1E8T) (78); parainfluenza virus 5, PIV5-HN (4JF7) (43); mumps virus, MuV-HN (5B2C) (79); Ghana virus, GhV-G (4UF7) (26); Nipah virus, NiV-G (2VWD) (39); Hendra virus, HeV-G (2X9M) (40); Cedar virus, CedV-G; measles virus, MeV-H (2RKC); and Mòjiāng virus, MojV-G (5NOP) (18). A pairwise evolutionary distance matrix was calculated using the Structural Homology Program (37) and plotted as an unrooted phylogenetic tree using PHYLIP (38).

### Cells and culture conditions

HEK-293T, Vero-CCL81, and U87 (American Type Culture Collection HTB-14) cells were grown in DMEM supplemented with 10% FBS and 1% penicillin/streptomycin (Thermo Fisher Scientific). Previously described CHO-pgsA745 cells expressing either wild-type or mutant ephrins (46) were maintained in DMEM/F12 medium (Thermo Fisher Scientific) with 10% FBS and 1% penicillin/streptomycin. HUVECs (pooled donors; Lonza) were maintained in EndoGRO-LS Complete Media Kit, composed of EndoGRO Basal Medium and the EndoGRO Supplement Kit (Millipore Sigma). EndGRO medium was also supplemented with 2% FBS, 1% penicillin/streptomycin, and 1% GlutaMAX supplement (Thermo Fisher Scientific). HUVECs used for soluble envelope competition experiments were between passage three and six.

### Plasmids and reagents

Codon-optimized sequences of CedV-G (GenBank accession number AJP33320.1) and CedV-F (GenBank accession number YP_009094085.1)

were tagged with a C-terminal HA or AU1, respectively, and cloned into pCAGGS mammalian expression vectors, as previously described for extant HNV glycoproteins ([26], [47]). Soluble human ephrin-B1–Fc, ephrin-B2–Fc, ephrin-B3–Fc, and Eph-B3–Fc used for competition experiments were purchased from R&D Systems.

## Cell surface binding assay

Binding of soluble Fc-tagged human ephrin-B ligands to HNV-G–transfected cells was assessed by flow cytometry. HEK-293T cells were transfected, using Lipofectamine 2000 (Thermo Fisher Scientific), with equal concentrations of plasmids encoding the indicated HNV-G proteins, or an empty vector control. Transfected cells were subsequently double-stained with a primary rabbit anti-HA polyclonal antibody (pAb) (Cat. no. NB600-363; Novus) at a 1:1,000 dilution, as well as increasing amounts of the ephrin-B1–Fc, ephrin-B2–Fc, and ephrin-B3–Fc ligands for 1 h at 4°C. Cells were then washed with buffer (2% FBS/PBS) and incubated with an Alexa488–labelled goat anti-rabbit antibody (Cat. no. A-11034; Thermo Fisher Scientific) diluted 1:2,000, to identify HNV-G positive cells, and an Alexa647–labelled goat anti-human antibody (Cat. no. A-21445; Thermo Fisher Scientific) diluted 1:2,000, to capture ephrin–B binding, for 1 h at 4°C. Cells were washed in 2% FBS/PBS, fixed in 2% PFA/PBS, and subjected to flow cytometry (Guava easyCyte).

Data were analyzed using FlowJo v10 software. Binding curves were generated by first gating on the live, HA-positive (i.e., HNV-G positive) population. The geometric mean fluorescence intensity (GMFI) in the red channel of this gated, G-positive population quantified the level of ephrin-B ligand binding. The highest GMFI value obtained within each dilution series was normalized to 100%, and cell surface $K_d$ values were calculated using GraphPad Prism.

## Pseudotyped virus

VSV particles pseudotyped with the HNV surface glycoproteins (HNVpp) were produced as previously described ([26], [47], [56]). Briefly, the particles were made from a VSV-ΔG-rLuc virus, a recombinant VSV derived from a full-length cDNA clone of the VSV Indiana serotype in which the VSV-G envelope protein is replaced by *Renilla* Luc. Pseudotyping was accomplished by transfecting HEK-293T cells (using BioT; Bioland Scientific) with expression plasmids containing the codon-optimized C-terminally tagged F and G envelope glycoproteins of NiV (NiVpp), HeV (HeVpp), GhV (GhVpp), CedV (CedVpp), or the VSV-G glycoprotein itself (VSVpp), and then infecting with VSV-ΔG-rLuc (complemented with VSV-G). Pseudotype-containing media were clarified 48 h after infection, by centrifugation at 200*g* for 5 min. Supernatants were then loaded on a 20% sucrose cushion and subject to ultra-centrifugation for 3 h at 110,000*g*. Concentrated pseudo-particle pellets were then resuspended in Dulbecco's PBS and stored at −80°C.

## Incorporation of henipaviral glycoproteins into HNVpp

Incorporation of F and G glycoproteins within NiVpp and CedVpp was determined by western blotting. Both F and G were detected using tag-specific anti-AU1 and anti-HA antibodies, respectively. Dilutions of NiVpp, CedVpp, and BALDpp (VSV virions bearing no

glycoprotein) were lysed in 6× Laemmli buffer (5% *β*-mercaptoethanol final), boiled for 10 min, and separated on an Any kD Mini-PROTEAN TGX Precast Protein Gel before transfer to a polyvinylidene difluoride (PVDF) membrane. Membranes were stained with a primary rabbit anti-HA pAb (Cat. no. NB600-363; Novus), diluted 1:2,000, and a primary rabbit anti-AU1 pAb (Cat. no. NB600-453; Novus), diluted 1:2,000. The membranes were then washed in PBS supplemented with 0.1% Tween-20 and incubated with Alexa647-labelled goat antirabbit antibody (Cat. no. A-21245; Thermo Fisher Scientific), diluted 1:4,000. To control for loading, the membranes were also stained for VSV matrix protein using a primary mouse anti-VSV-M mAb (Cat. no. EB0011; Kerafast), diluted 1:1,1000, and a secondary Alexa647-labelled goat anti-mouse antibody (Cat. no. A-21236; Thermo Fisher Scientific), diluted 1:3,000. Membranes were imaged using a Bio-Rad ChemiDoc and densitometry analysis was performed by using the Bio-Rad Image Lab 5.1 software.

## Quantification of viral genome copies

Viral RNA was extracted from HNVpp preparations using the QIAamp viral RNA Mini Kit (QIAGEN) and subsequently reverse-transcribed using the Tetro cDNA Synthesis Kit (Bioline). VSV genome copy number was quantified by qPCR, using the SensiFAST SYBR & Fluorescein kit (Bioline), using genome-specific primers against the VSV Indiana L region (sequences available on request). Standard curves were generated by a serial dilution of the full-length VSV-ΔG-rLuc genomic plasmid.

## Quantification of viral entry

Target cells were grown in 96-well plates and infected with the pseudoviruses serially diluted in appropriate cell culture medium. For soluble ephrin-B entry inhibition experiments, the indicated amounts of soluble ephrin-B–Fc (R&D Systems) were incubated together with a pseudotyped virus for 1 h at 37°C. The mixture of virus and soluble ephrin-B was then added to the target cells. For soluble envelope entry inhibition experiments, the indicated quantity of soluble HNV-G–Fc (production described above) or soluble Eph-B3–Fc (R&D Systems) were incubated with the adherent target cells for 1 h at 37°C. After incubation with soluble protein, pseudotyped virus was added. To assess viral entry into the wild-type and mutant ephrin-expressing cell lines ([Fig 6D]), equivalent amounts of NiVpp and CedVpp (titrated to give roughly equivalent rLuc activities on CHO-B2 cells) were used to infect the distinct ephrin-expressing CHO-pgsA745 cell lines.

In all instances, the quantity of pseudotyped virus stock used was predetermined to fall within the linear dynamic range of *Renilla* Luc detection. Furthermore, in all HNVpp entry experiments, infected cells were washed with PBS and lysed 24 h postinfection. Cell lysates were subsequently processed using a *Renilla* Luciferase Detection Kit, according to the manufacturer's directions (Promega). Luminescence intensity was measured using a Cytation3 Plate Reader.

## Quantification of ephrin-B mRNA transcripts in primary HUVECs

Ephrin-B1, -B2, and -B3 mRNA transcripts present in the primary HUVECs were quantified using qPCR. Total RNA was extracted from

$3 \times 10^5$ HUVECs with a NucleoSpin RNA isolation kit (Macherey-Nagel). Processed mRNA transcripts were reverse-transcribed into cDNA using oligo(dT) primers and the Tetro cDNA synthesis kit (Bioline). qPCR was performed using the SensiFAST SYBR & Fluorescein Kit (Bioline), using gene-specific primers (sequences available on request) for ephrin-B1, -B2, and -B3. Standard curves for each gene were generated by a serial dilution of the ephrin-B pcDNA3.1 expression plasmids. As a normalization control, hypoxanthine phosphoribosyltransferase (HPRT) copy numbers were also determined.

### HNV glycoprotein-mediated syncytia formation in U87 glioblastoma cells

Syncytia assays were performed by transfecting U87 cells with HNV-F and HNV-G expression plasmids or an empty vector control (using Lipofectamine 2000; Thermo Fisher Scientific). Bright-field images were taken at 48 h posttransfection with a Nikon Eclipse TE300 Diaphot Microscope. Images represent two random fields for each condition.

### Data deposition

The atomic coordinates and structure factors for CedV-G and CedV-G–ephrin-B1 have been deposited in the PDB with the accession codes 6THB and 6THG, respectively.

## Supplementary Information

## Acknowledgements

We are grateful to Diamond Light Source for beamtime (proposal MX19946), the staff of beamlines I02 and I24 for assistance with data collection, and to Helen M Ginn for helpful discussions. We thank the Medical Research Council (MR/L009528/1 and MR/S007555/1 to TA Bowden), Academy of Finland (#309605 to I Rissanen), and National Institutes of Health (NIH) (National Institute of Allergy and Infectious Diseases [NIAID] AI123449, AI069317, and AI115226 to B Lee) for funding. K Azarm was supported by the National Institute of Allergy and Infectious Diseases of the NIH under Award Number F31-AI133943 & the Host-Pathogens Interactions Training Grant T32-AI007647-16 at the Icahn School of Medicine at Mount Sinai. B Lee also acknowledges the Ward Coleman estate for endowing the Ward-Coleman Chairs at the Icahn School of Medicine at Mount Sinai. The Wellcome Centre for Human Genetics is supported by Wellcome Centre grant 203141/Z/16/Z. The Genotype-Tissue Expression (GTEx) Project was supported by the Common Fund of the Office of the Director of the NIH and by other relevant institutes indicated on the GTEx website. The data used for the analyses described in this manuscript (Fig S5) were obtained from the GTEx Portal dbGaP accession number phs000424.v7.p2 on July 24, 2019.

### Author Contributions

R Pryce: conceptualization, data curation, formal analysis, investigation, methodology, and writing—original draft, review, and editing.

K Azarm: conceptualization, data curation, formal analysis, investigation, methodology, and writing—original draft, review, and editing.

I Rissanen: conceptualization, data curation, formal analysis, investigation, methodology, and writing—review and editing.

K Harlos: resources, methodology, and data curation.

TA Bowden: conceptualization, formal analysis, supervision, funding acquisition, methodology, and writing—original draft, review, and editing.

B Lee: conceptualization, formal analysis, supervision, funding acquisition, methodology, and writing—original draft, review, and editing.

### Conflict of Interest Statement

The authors declare that they have no conflict of interest.

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
