## [Reviewer comments · Life Science Alliance]

Life Science Alliance

A key region of molecular specificity orchestrates unique ephrin-B1 utilization by Cedar virus

Rhys Pryce, Kristopher Azarm, Ilona Rissanen, Karl Harlos, Thomas Bowden, and Benhur Lee
DOI: <https://doi.org/10.26508/lsa.201900578>

Corresponding author(s): Benhur Lee, Icahn School of Medicine at Mount Sinai and Thomas Bowden, University of Oxford

Review Timeline:

Submission Date:	2019-10-16
Editorial Decision:	2019-11-13
Revision Received:	2019-11-25
Editorial Decision:	2019-11-27
Revision Received:	2019-12-03
Accepted:	2019-12-04

Scientific Editor: Andrea Leibfried

Transaction Report:

November 13, 2019

Re: Life Science Alliance manuscript #LSA-2019-00578-T

Dr. Benhur Lee
Icahn School of Medicine at Mount Sinai
New York, NY 10029

Dear Dr. Lee,

Thank you for submitting your manuscript entitled "A key region of molecular specificity orchestrates unique ephrin-B1 utilization by Cedar virus" to Life Science Alliance. The manuscript was assessed by expert reviewers, whose comments are appended to this letter.

As you will see, the reviewers note the large overlap of your conclusions with previous work. Since the related study only recently got published, this is not precluding publication here. We would thus like to invite you to submit a revised version of your manuscript for publication in Life Science Alliance. Both reviewers provide constructive input on how to further strengthen your work, and the revision requests seem straightforward to address. Please do get in touch in case you would like to discuss individual revision points further.

Thank you for this interesting contribution to Life Science Alliance. We are looking forward to receiving your revised manuscript.

Sincerely,

Andrea Leibfried, PhD
Executive Editor

Life Science Alliance
Meyerhofstr. 1
69117 Heidelberg, Germany
t +49 6221 8891 502
e a.leibfried@life-science-alliance.org
www.life-science-alliance.org

B. MANUSCRIPT ORGANIZATION AND FORMATTING:

Reviewer #1 (Comments to the Authors (Required)):

In the manuscript, Rhys et al. report CedV could use not only ephrin-B2, which is a common HNV receptor, but also ephrin-B1 as an entry receptor. They determined the crystal structure of CedV-G protein at a resolution of 2.78-Å, and the CedV -G bound to ephrin-B1 complex structure at a resolution of 4.07-Å. Structural analyses reveal that diverse HNV-G proteins bind to their distinct ephrin receptors in a conserved binding mode, while subtle structural features of CedV contribute to its unique ephrin ligand specificity.

Overall, the paper is written clearly, and the findings represent an important advancement for HNV viral entry. However, the major concern is the novelty. There is a paper published on PNAS recently, titled "Structural and functional analyses reveal promiscuous and species specific use of ephrin receptors by Cedar virus". They demonstrate that CedV can use ephrin-B1, A2, A5 to enter cells, and determined the CedV-G structure at 3.7-Å resolution, and the complex structures of CedV-G with ephrin-B1 or B2 at 3.5-Å and 2.85-Å respectively. It seems this PNAS paper has more data, especially they have the exact affinity data of the protein binding . Both two studies have similar conclusions.

Minor concerns: In Figure 6, it is better to add a panel showing the sequence alignment of different HNV-G, to highlight the critical binding sites.

Reviewer #2 (Comments to the Authors (Required)):

In this manuscript, Pryce et al demonstrate that EphrinB1, in addition to EphrinB2, but not EphrinB3, is a functional receptor for CedV - a member of the Henipavirus (HNV) genus. The data are supported by comprehensive structural/biochemical experiments including binding assay, pseudovirus entry assay and structural analysis. Utilization of EphrinB1 as an entry receptor has not been reported for any members of the HNV genus before this paper and a recently published paper by Laing et al (PMID: 31548390). This is an important study as it reveals the ability of HNVs to use different members of the Ephrin protein family for entry and shed light on the molecular barriers that dictate specific receptor usage by different HNVs. The data are well presented and the discussion section is thorough.

This reviewer has several comments/suggestions described below:

In the 2nd part of the Results section, since the authors use ephrin constructs that are fused to Fc, their dimeric nature leads to avidity and artificially enhance the affinity. Thus, these are not genuine Kds.

The authors should cite and discuss the recently reported findings by Liang et al (PMID: 31548390) (including comparing their structure to theirs).

Reported buried surface area (BSA): the authors should clarify that the values in this manuscript are total BSA on EphrinB1 and CedV G (to be consistent with Liang et al, which reported the values on one interface only).

The authors state "Together, binding-induced structural transitions within both CedV-G and the ephrin G-H loops support a model of an induced-fit mechanism of ephrin recognition that is

conserved across ephrin-tropic HNVs ".

How did the authors distinguish between true induced-fit and selection from a conformational equilibrium?

"Similarly, both ephrin-B2- Fc and ephrin-B1-Fc inhibited CedVpp entry into CHO-B2 cells (Fig. 5d, right panel), evidencing the ability of ephrin-B1 to block ephrin-B2-dependent CedVpp entry through competition for an overlapping binding site on CedV-G. Moreover, ephrin-B2-Fc inhibited CedVpp entry into CHO-B1 cells (Fig. 5d, left panel). In both CHO-B2 and CHO-B1 cells, ephrin-B2-Fc-mediated inhibition of CedV-G was more potent than ephrin-B1-Fc (Fig. 5d), further supporting our binding (Fig. 2) and entry (Fig. 3) data that suggest ephrin-B2 is more efficiently utilized than ephrin-B1." Are the structural data consistent with the authors' claim that CedV G utilize EphrinB2 more efficiently than it does EphrinB1? Or what could rationalize this preferred utilization? Also, these results confirm the recently reported structures by Laing et al (PMID 31548390) and this should be mentioned.

How do the authors reconcile the virtually identical binding responses of NiV G to ephrin-B2 or ephrin-B3 (Fig 2 a) with the 3 log difference observed between ephrin-B2 or ephrin-B3 for inhibition of NiVpp entry in CHO-B2 cells (Fig 5 d)?

"Although CedV-G is unable to utilize ephrin-B3, our structural hypothesis suggests that acquired ephrin-B1 specificity does not necessarily come at the expense of ephrin-B3 usage, as the LW motif is common to both ephrin-B2 and ephrin-B3."

Could the authors use their structural data to explain why Ephrin B3 is incompatible with CedV G?

Could the authors discuss how conserved EphrinB1 is among different species and whether CedV can transmit among species by utilizing EphrinB1?

"As expected [43, 54, 55], ephrin- B2-Fc and ephrin-B3-Fc inhibited NiVpp entry into CHO-B2 cells, while ephrin-B1-Fc failed to strongly inhibit entry at concentrations as high as 10 nM (Fig. 5d, middle panel), confirming that ephrin-B2 and ephrin-B3 are each bound by the same site on NiV-G [45] "

Please cite PMID 18632560 which reports the structure of NiV G with ephrin-B3.

The Editorial Board
Life Science Alliance

Dear Dr. Leibfried,

We wish to submit the revised manuscript by Pryce and Azarm *et al.* entitled 'A key region of molecular specificity orchestrates unique ephrin-B1 utilization by Cedar virus' to be considered for publication in *Life Science Alliance*. We thank the reviewers for their responses, and below we address the comments made point by point (our responses in blue and changes to the text in red).

Reviewer #1:

In the manuscript, Rhys et al. report CedV could use not only ephrin-B2, which is a common HNV receptor, but also ephrin-B1 as an entry receptor. They determined the crystal structure of CedV-G protein at a resolution of 2.78-Å, and the CedV -G bound to ephrin-B1 complex structure at a resolution of 4.07-Å. Structural analyses reveal that diverse HNV-G proteins bind to their distinct ephrin receptors in a conserved binding mode, while subtle structural features of CedV contribute to its unique ephrin ligand specificity.

Overall, the paper is written clearly, and the findings represent an important advancement for HNV viral entry. However, the major concern is the novelty. There is a paper published on PNAS recently, titled "Structural and functional analyses reveal promiscuous and species specific use of ephrin receptors by Cedar virus". They demonstrate that CedV can use ephrin-B1, A2, A5 to enter cells, and determined the CedV-G structure at 3.7-Å resolution, and the complex structures of CedV-G with ephrin-B1 or B2 at 3.5-Å and 2.85-Å respectively. It seems this PNAS paper has more data, especially they have the exact affinity data of the protein binding. Both two studies have similar conclusions.

Response: We thank the Reviewer for their efficient capsulation of our study. This manuscript, in its exact form, was deposited into bioRxiv *before* the PNAS study was published, or even available online. This journal has editorial policies that do not make prior publications a consideration in evaluating manuscripts under their scoop protection policies. We believe in the open and transparent peer review system that *Life Sciences Alliance* follow. We agree with the reviewer that any comparisons with the *PNAS* study by Laing et al. should and can be made in open peer review forums.

Minor concerns: In Figure 6, it is better to add a panel showing the sequence alignment of different HNV-G, to highlight the critical binding sites.

Response: We completely agree with the reviewer that providing the annotated sequence alignment of HNV-G proteins will help add clarity and give additional context to Fig. 6. Indeed, such an annotated sequence alignment with additional structural annotations was provided in Supplementary Figure 1. We felt these data were best presented as an independent figure, given (1) the discontinuous nature of the ephrin-binding site, (2) the size of the resulting sequence alignment, and (3) the additional information content provided by the structural annotations (e.g. contact residues, disulphide bonds, and N-linked glycan sites etc.)

Reviewer #2:

In this manuscript, Pryce et al demonstrate that EphrinB1, in addition to EphrinB2, but not EphrinB3, is a functional receptor for CedV - a member of the Henipavirus (HNV) genus. The data are supported by comprehensive structural/biochemical experiments including binding assay, pseudovirus entry assay and structural analysis. Utilization of EphrinB1 as an entry receptor has not been reported for any members of the HNV genus before this paper and a recently published paper by Laing et al (PMID: 31548390). This is an important study as it reveals the ability of HNVs to use different members of the Ephrin protein family for entry and shed light on the molecular barriers that dictate specific receptor usage by different HNVs. The data are well presented and the discussion section is thorough.

This Reviewer has several comments/suggestions described below:

In the 2nd part of the Results section, since the authors use ephrin constructs that are fused to Fc, their dimeric nature leads to avidity and artificially enhance the affinity. Thus, these are not genuine K_d s.

Response: We thank the Reviewer for noting this point and allowing us to clarify the text. We agree that avidity effects resulting from Fc-tagged HNV-G proteins may yield avidity enhanced binding and thus different estimates of K_d relative to experiments utilizing monomeric proteins. As such, we do not undertake direct quantitative comparison with other studies quoting bimolecular interaction kinetic values. We instead present values for all ephrin-tropic HNV-Gs, to permit quantitative comparison of all HNV-G ephrin pairs utilizing a uniform experimental set-up. To avoid any confusion, we have updated text in this section:

Line 159: The section title has been updated to remove the phrase ‘nanomolar affinity’ and now reads ‘CedV-G binds both ephrin-B1 and ephrin-B2’.

Line 163–165: Human embryonic kidney (HEK) 293T cells were transfected with HA-tagged HNV-G glycoproteins (NiV-G, HeV-G, GhV-G, and CedV-G) and titrated against soluble Fc-tagged human B-type ephrins (ephrin-B1-Fc, ephrin-B2-Fc, and ephrin-B3-Fc) to obtain apparent dissociation constants (K_d).

The authors should cite and discuss the recently reported findings by Laing et al (PMID: 31548390) (including comparing their structure to theirs).

Response: Since this manuscript was deposited into BioRxiv before the Laing *et al.* study was published, even online, we were not able to make any reference to it. In line with editorial advice, we now cite the Laing *et al.* (PMID: 31548390) reference as an **endnote** following the **Acknowledgements** section.

“Endnote: While the preprint version of this paper [<https://doi.org/10.1101/724138>] was under peer review, Laing et al. published a study (PMC6789926) reporting similar results.”

Reported buried surface area (BSA): the authors should clarify that the values in this manuscript are total BSA on EphrinB1 and CedV G (to be consistent with Laing et al, which reported the values on one interface only).

Response: We have updated the text in order to clarify that the values are total BSA for both components of the complex, which is consistent with previous studies of HNV-G–ephrin complexes (Bowden et al., 2008; Xu et al., 2008; Lee et al., 2015):

Line 298–302: CedV-G and ephrin-B1 form a 1:1 complex with an extensive molecular interface that buries a combined surface area of 2,900 Å² (1,450 Å² per component). The buried interface is larger than that previously characterized for other HNV-G–ephrin complexes (NiV-G–ephrin-B2 = 2,800 Å²; NiV-G–ephrin-B3 = 2,700 Å²; HeV-G–ephrin-B2 = 2,600 Å²; GhV-G–ephrin-B2 = 2,300 Å², calculated using the PDBePISA server [54]).

Line 303-305: (CedV-) G, which contribute ~70% (~2,000 Å²) of the total buried surface area (BSA; Figs. 1, 5c, and Supplementary Fig. 1).

The authors state "Together, binding-induced structural transitions within both CedV-G and the ephrin G-H loops support a model of an induced-fit mechanism of ephrin recognition that is conserved across ephrin-tropic HNVs".

How did the authors distinguish between true induced-fit and selection from a conformational equilibrium?

Response: We thank the Reviewer for bringing up this important point and agree the presented structures do not provide grounds to distinguish induced fit from conformational selection. As such, we have altered the text as follows:

Line 314–316: Differences between the structural states of both CedV-G and ephrin-B1 may represent an induced-fit mechanism of ephrin recognition, which has been postulated for other ephrin-tropic HNV-Gs [36, 37, 45], or selection from a conformational equilibrium.

Furthermore, wording elsewhere has been altered for the purpose of clarity:

Line 294–295: Structural plasticity within this region is observed in other HNV-G proteins and their ephrin-bound complexes [36, 37].

"Similarly, both ephrin-B2-Fc and ephrin-B1-Fc inhibited CedVpp entry into CHO-B2 cells (Fig. 5d, right panel), evidencing the ability of ephrin-B1 to block ephrin-B2-dependent CedVpp entry through competition for an overlapping binding site on CedV-G. Moreover, ephrin-B2-Fc inhibited CedVpp entry into CHO-B1 cells (Fig. 5d, left panel). In both CHO-B2 and CHO-B1 cells, ephrin-B2-Fc-mediated inhibition of CedV-G was more potent than ephrin-B1-Fc (Fig. 5d), further supporting our binding (Fig. 2) and entry (Fig. 3) data that suggest ephrin-B2 is more efficiently utilized than ephrin-B1." Are the structural data consistent with the authors' claim that CedV G utilize EphrinB2 more efficiently than it does EphrinB1? Or what could rationalize this preferred utilization? Also, these results confirm the recently reported structures by Laing et al (PMID 31548390) and this should be mentioned.

How do the authors reconcile the virtually identical binding responses of NiV G to ephrin-B2 or ephrin-B3 (Fig 2 a) with the 3 log difference observed between ephrin-B2 or ephrin-B3 for inhibition of NiVpp entry in CHO-B2 cells (Fig 5 d)?

Response: We thank the reviewer for noting this 2-log (not 3-log) difference. Indeed, we also detect a similar difference in the ephrin-B2 and ephrin-B3 IC₅₀ values of NiVpp entry inhibition on the Vero-CCL81s (Figure 2b). We have previously noted this apparent discrepancy (e.g. Ref 46: Negrete, O.A. et al., *PLoS Pathog*, 2006; compare Figures 1B & 3), which is a known observation that has repeatedly appeared in the literature. Envelope-receptor interactions that lead to binding versus inhibition of entry

measure two different things. Binding measures direct protein-protein interactions under equilibrium conditions, whereas entry involves a host of allosteric signals from receptor binding, to F-triggering, to fusion-pore formation, which then results in delivery of the viral RNP into the host cell cytoplasm, and subsequent transcription and viral genome replication that leads to detection of the reporter gene signal. Thus, small differences in binding affinity, and more importantly, differences in the *efficiency* of how particular receptors (e.g. ephrin-B2 versus ephrin-B3) allosterically trigger F proteins (not measured by receptor binding affinities alone) can amplify any putative differences in envelop-receptor interactions that result in entry. How different receptors mechanistically trigger productive fusion and entry by various paramyxoviruses is a subject of intense investigation by aficionados of paramyxovirus entry and is beyond the scope of this current study.

"Although CedV-G is unable to utilize ephrin-B3, our structural hypothesis suggests that acquired ephrin-B1 specificity does not necessarily come at the expense of ephrin-B3 usage, as the LW motif is common to both ephrin-B2 and ephrin-B3."

Could the authors use their structural data to explain why Ephrin B3 is incompatible with CedV G?

Response: We thank the Reviewer for the opportunity to clarify this point. The precise structural determinants that prevent recognition of ephrin-B3 by CedV-G are presently unclear. Indeed, answering such questions is a focus of ongoing work. We have updated the Discussion to highlight this uncertainty rather than propose speculative hypotheses:

Line 456–459: Whilst the molecular features that preclude ephrin-B3 utilization by CedV-G remain unclear, our structural hypothesis suggests that acquired ephrin-B1 specificity does not necessarily come at the expense of ephrin-B3 usage, as the LW motif is common to both ephrin-B2 and ephrin-B3.

Could the authors discuss how conserved EphrinB1 is among different species and whether CedV can transmit among species by utilizing EphrinB1?

Response: Ephrin-B1 is highly conserved amongst mammalian species (96-99% sequence similarity), at least equal to if not slightly less so than ephrin-B2. We have added in a statement with regards to ephrin-B1 conservation in the relevant part of the Discussion section:

Line 472-475: Of note is the relatively high expression of ephrin-B1 in the lung, esophagus, and salivary glands (Supplementary Fig. 5), which suggests that ephrin-B1 utilization could augment aspects of oropharyngeal transmission postulated for HNV [66], especially since ephrin-B1 is almost as conserved as ephrin-B2 across mammalian species (96-99% sequence similarity).

"As expected [43, 54, 55], ephrin-B2-Fc and ephrin-B3-Fc inhibited NiVpp entry into CHO-B2 cells, while ephrin-B1-Fc failed to strongly inhibit entry at concentrations as high as 10 nM (Fig. 5d, middle panel), confirming that ephrin-B2 and ephrin-B3 are each bound by the same site on NiV-G [45]"

Please cite PMID 18632560 which reports the structure of NiV G with ephrin-B3.

Response: We have added the appropriate citation (ref 48), as requested:

Line 345–349: To assess this, we determined pseudotyped virus entry into CHO-B1 and CHO-B2 cells in the presence of competing soluble B-class ephrin ligands. As expected [46, 56, 57], ephrin-

B2-Fc and ephrin-B3-Fc inhibited NiVpp entry into CHO-B2 cells, while ephrin-B1-Fc failed to strongly inhibit entry at concentrations as high as 10 nM (Fig. 5d, middle panel), confirming that ephrin-B2 and ephrin-B3 are each bound by the same site on NiV-G [43, 48].

November 27, 2019

RE: Life Science Alliance Manuscript #LSA-2019-00578-TR

Dr. Benhur Lee
Icahn School of Medicine at Mount Sinai
Microbiology
One Gustave L. Levy Place
#1124
New York, NY 10029

Dear Dr. Lee,

Thank you for submitting your revised manuscript entitled "A key region of molecular specificity orchestrates unique ephrin-B1 utilization by Cedar virus". I appreciate the introduced changes and would be happy to publish your paper in Life Science Alliance pending final revisions necessary to meet our formatting guidelines:

- I think it is better to include the current endnote in the actual discussion. You could do so in the following way:

[...] Thus, whilst the pathobiological and ecological implications of ephrin-B1 tropism are presently unclear, our study sets a precedent for ephrin-B1 utilization, and in so doing expands the known repertoire of HNV cellular entry receptors utilized by this group of lethal human pathogens. While the preprint version of this paper [preprint; 67] was under peer review, Laing et al. published a study reporting similar results [68]. [...]

Please also add the citations to the preprint and to Laing et al in your reference list.

- Please add a callout in the manuscript text to Fig S4
- Please add information on Fig 4C in the legend to this figure
- Please list 10 authors et al in the reference list
- Please add a scale bar to Fig S3b
- The inset in Fig S3b does not match the zoomed area; please fix

To avoid unnecessary delays in the acceptance and publication of your paper, please read the

following information carefully.

A. FINAL FILES:

B. MANUSCRIPT ORGANIZATION AND FORMATTING:

Sincerely,

December 4, 2019

RE: Life Science Alliance Manuscript #LSA-2019-00578-TRR

Dr. Benhur Lee
Icahn School of Medicine at Mount Sinai
Microbiology
One Gustave L. Levy Place
#1124
New York, NY 10029

Dear Dr. Lee,

Thank you for submitting your Research Article entitled "A key region of molecular specificity orchestrates unique ephrin-B1 utilization by Cedar virus". It is a pleasure to let you know that your manuscript is now accepted for publication in Life Science Alliance. Congratulations on this interesting work.

DISTRIBUTION OF MATERIALS:

Again, congratulations on a very nice paper. I hope you found the review process to be constructive and are pleased with how the manuscript was handled editorially. We look forward to future exciting submissions from your lab.

Sincerely,

Andrea Leibfried, PhD
Executive Editor
Life Science Alliance
Meyerohofstr. 1
69117 Heidelberg, Germany
t +49 6221 8891 502
e a.leibfried@life-science-alliance.org
www.life-science-alliance.org